

# Color components determination and full-length comparative transcriptomic analyses reveal the potential mechanism of carotenoid synthesis during *Paphiopedilum armeniacum* flowering

Yiwei Bai[1,2,*], Jiping Ma[1,2,3,*], Yanjun Ma[1,2,4], Yanting Chang[1,2], Wenbo Zhang[1,2,4], Yayun Deng[1,2], Na Zhang[1,2], Xue Zhang[1,2], Keke Fan[1,2], Xiaomeng Hu[1,2], Shuhua Wang[1,2], Zehui Jiang[1,2] and Tao Hu[1,2,4]

[1] International Center for Bamboo and Rattan, Beijing, China
[2] Key Laboratory of National Forestry and Grassland Administration/Beijing for Bamboo & Rattan Science and Technology, Beijing, China
[3] China Forestry Publishing House, Xicheng District, Beijing, China
[4] Pingxiang Bamboo Forest Ecosystem Research Station, Pingxiang, China
[*] These authors contributed equally to this work.

Corresponding author
Tao Hu, hutao@icbr.ac.cn

## ABSTRACT

**Background**. *Paphiopedilum armeniacum* (*P. armeniacum*), an ornamental plant native to China, is known for its distinctive yellow blossoms. However, the mechanisms underlying *P. armeniacum* flower coloration remain unclear.

**Methods**. We selected *P. armeniacum* samples from different flowering stages and conducted rigorous physicochemical analyses. The specimens were differentiated based on their chemical properties, specifically their solubilities in polar solvents. This key step enabled us to identify the main metabolite of flower color development of *P. armeniacum*, and to complete the identification by High-performance liquid chromatography (HPLC) based on the results. Additionally, we employed a combined approach, integrating both third-generation full-length transcriptome sequencing and second-generation high-throughput transcriptome sequencing, to comprehensively explore the molecular components involved.

**Results**. We combined physical and chemical analysis with transcriptome sequencing to reveal that carotenoid is the main pigment of *P. armeniacum* flower color. Extraction colorimetric method and HPLC were used to explore the characteristics of carotenoid accumulation during flowering. We identified 28 differentially expressed carotenoid biosynthesis genes throughout the flowering process, validated their expression through fluorescence quantification, and discovered 19 potential positive regulators involved in carotenoid synthesis. Among these candidates, three *RCP2* genes showed a strong potential for governing the *PDS* and *ZDS* gene families. In summary, our study elucidates the fundamental mechanisms governing carotenoid synthesis during *P. armeniacum* flowering, enhancing our understanding of this process and providing a foundation for future research on the molecular mechanisms driving *P. armeniacum* flowering.

# INTRODUCTION

*Paphiopedilum armeniacum* (*P. armeniacum*), a perennial herb endemic to China, specifically in the counties of Fugong and Lushui in Yunnan Province, possesses an additional distribution range extending into Myanmar. This remarkable species has suffered a precipitous decline in population size due to both the serious destruction of wild resources and the degradation of natural habitats caused by natural and anthropogenic factors. Consequently, *P. armeniacum* teetered onto the brink of extinction. Accordingly, it was listed in the Convention on International Trade in Endangered Species of Wild Fauna and Flora (*Cao et al., 2022*; *Wang et al., 2021*). *P. armeniacum* boasts exceptional ornamental value, epitomized by the rarity as a yellow-flower member within the Paphiopedilum. It has large and elegant flowers, a long flowering period, and beautiful markings on its leaves. It won a gold medal many times in the World Orchid Show. The petals of *P. armeniacum* exhibit a predominantly apricot-yellow hue almost as a whole, rendering it a crucial resource for breeding endeavors and serving as fundamental research material (*Mou et al., 2012*). The cultivation of flower color holds significant prominence in ornamental flower breeding, as alterations in flower coloration have the potential to enhance both ornamental value and economic returns (*Morimoto et al., 2021*). Therefore, exploring the mechanism of flower color formation in *P. armeniacum* demonstrates a certain representative significance and can be applied as a crucial part of comprehending the intricate network of yellow flower color synthesis in Paphiopedilum.

Determination of plant flower color predominantly depends on three principal classes of compounds: flavonoids, carotenoids, and alkaloids (*Młodzińska, 2009*; *Tanaka, Sasaki & Ohmiya, 2008*). Alkaloids, for the most part, lack inherent coloration, whereas a limited subset of these compounds possess chromogenic properties and are primarily distributed among Caryophyllaceae plants (*Clement & Mabry, 1996*). Therefore, in most plant species, flavonoids and carotenoids are the main pigments that determine flower coloration (*Tanaka, Sasaki & Ohmiya, 2008*; *Zhao & Tao, 2015*). Flavonoids have received extensive scrutiny as plant secondary metabolites, with culmination of their metabolic pathways predominantly leading to the production of anthocyanins and co-pigments. Anthocyanins are mainly exist in the vacuoles of water-soluble pigment (*Clement & Mabry, 1996*; *Weiss, 1995*), have been demonstrated to be jasmone acid methyl ester (MeJA) induced synthesis (*Zhang et al., 2023*). Previous studies have indicated that flavonoids are the main source of color for the petals of yellow flowers, such as *Lonicera japonica Thunb* and *Eustoma grandiflorum* (*Fossen & Ovstedal, 2003*; *Markham, Gould & Ryan, 2001*; *Zhou et al., 2013*). Carotenoids, a general term for carotene and its oxidized derivative lutein, represent a class of highly unsaturated compounds (polyenes). These fat-solute pigments predominantly

reside within the plastids (*Sun et al., 2018*). Studies have demonstrated that carotenoids play a central role in the correlation between yellow oncidium and *chrysanthemum* (*Hattori, 1991*; *Hieber, Mudalige-Jayawickrama & Kuehnle, 2005*). Although anthocyanins are crucial pigments in the determination of flower color in most plants, carotenoids continue to exert a significant influence on petal coloration in yellow flowers (*Iwashina, 2015*; *Markham, Gould & Ryan, 2001*). Therefore, the study of the petal coloration mechanism of *P. armeniacum* is inseparable from the discussion of petal coloration components.

In recent years, advances in molecular biology have propelled the advancement of research pertaining to flower color-related genes, leading to increasing levels of maturity in this field (*Zhao & Tao, 2015*; *Zhang, Butelli & Martin, 2014*). Taking the carotenoid biosynthesis pathway as an example, research on genes related to the carotenoid biosynthesis pathway has matured. The highly conserved carotenoid biosynthesis pathway (CBP) has been characterized and verified in numerous plant species (*Meier et al., 2011*; *Ohmiya et al., 2019*; *Ruiz-Sola & Rodríguez-Concepción, 2012*). Currently, the focus has shifted towards unraveling the multifaceted regulatory mechanisms of carotenoid accumulation in fruits and leaves, while studies elucidating these mechanisms in flowers remain scarce (*Zhu et al., 2010*).

Recently, *Li et al. (2023)* conducted an in-depth exploration of the formation mechanism of yellow flowers in rape, elucidated the molecular mechanism of carotenoid storage and yellow flower formation in rape, and provided novel insights and research perspectives for studying the relationship between carotenoids and yellow flowers. It is noteworthy that the regulatory factors involved in tomato fruit ripening exert minimal influence on petal coloration, which may indicate the existence of a distinct and intricate regulatory network governing carotenoid synthesis specifically in flowers (*Stanley & Yuan, 2019*). Among the transcription factors found up to date, only *MYB305, RCP1*, and *RCP2* have been confirmed as regulators of carotenoid synthesis during flowering. *MYB305* may mediate transcriptional regulation of *NtPSY, NtZDS*, and *NtLCY*. RNAi knockdown of *MYB305* causes the loss of $\beta$-carotene in floral nectaries. *RCP1* and *RCP2* are required for carotenoid biosynthesis in petals and positively regulate almost all CBP genes expressed in the flowers (*Liu et al., 2009*; *Sagawa et al., 2015*; *Stanley et al., 2020*; *Wang et al., 2014a*; *Wang et al., 2014b*).

The primary objective of this study was to investigate the underlying factors contributing to yellow petal coloration in *P. armeniacum* flowers through the application of differences in chemical properties between carotenoids and flavonoids. To achieve this, floral organs at three different stages throughout the flowering process were selected as samples. Using transcriptome and full-length transcriptome sequencing techniques, we aimed to elucidate the crucial structural genes and associated transcription factors involved in the intricate process of flower color formation in *P. armeniacum*. This study delved into the mechanisms underlying petal coloration of *P. armeniacum* to a certain extent, thus providing valuable insights into the study of the flowering mechanism of *P. armeniacum* flowers.

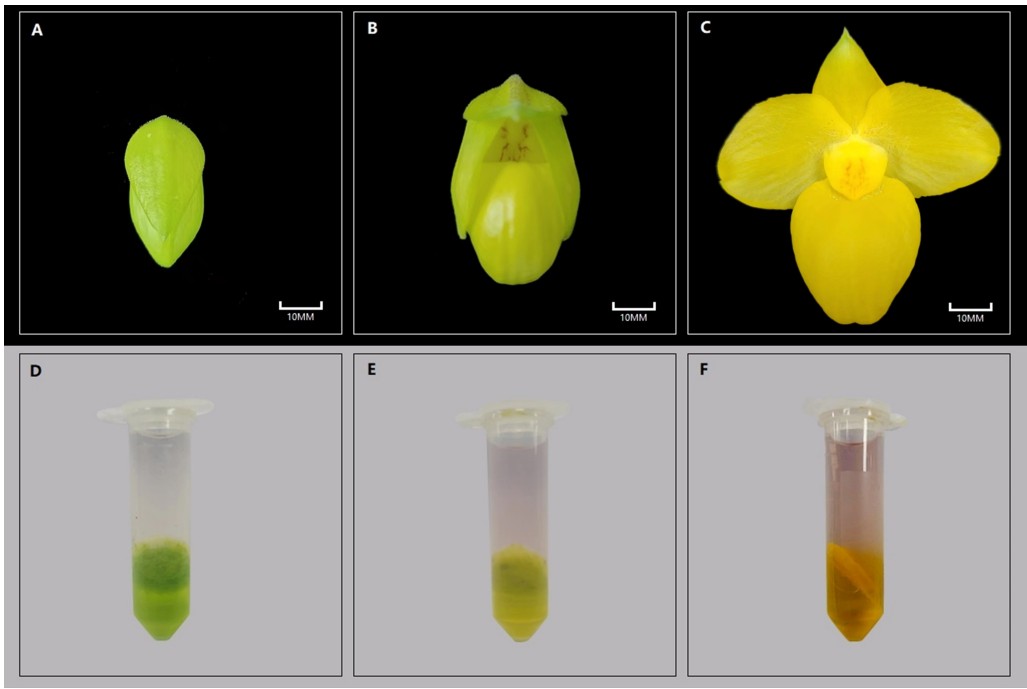

**Figure 1** **Color results of *P. armeniacum* flower organs and reagent delamination experiments at different stages.** Color results of *P. armeniacum* flower organs and reagent delamination experiments at different stages. (A–C) Flower bud stage, initial flowering stage, and full flowering stage, respectively. (D–E) Delaminated color results of reagents in the aforementioned stages.

# MATERIALS & METHODS

## Materials

The *P. armeniacum* specimens utilized in the experiment were introduced into Xingyi, Guizhou, and cultivated in a controlled greenhouse at Luyuan Flower Co., Ltd. To capture the distinct stages of the flowering process, three specific developmental stages were selected as follows: the flower bud stage (the flower buds were in a yellow-green unbloomed state), the initial flowering stage (the flower buds were not fully opened, and the petals were a little green), and the full flowering stage (the buds were fully open, and the overall appearance was apricot yellow) (Fig. 1). These three developmental stages are the most representative developmental nodes in the flowering process and have good differentiation to facilitate sample collection. Within each stage, four individuals demonstrating similar growth levels were selected for sampling, which ensured the subsequent completion of transcriptome sequencing, carotenoid quantitative analysis, and real-time quantitative polymerase chain reaction (RT-qPCR) analysis.

## Methods

### Color component analysis

According to the differences in the physical and chemical properties of flavonoids and carotenoids, a chemical reagent analysis was adopted to detect the origin of the yellow

coloration in *P. armeniacum* flowers (*Amorim-Carrilho et al., 2014*). Flowers at different developmental stages, including the bud stage, initial flowering stage, and full flowering stage, were selected as materials to ensure their health, consistent growth, and uniform coloration. Subsequently, the selected materials were ground into fine powders using liquid nitrogen, enveloped in 500 μl methanol, and s centrifuged at room temperature (12000 rpm for 5 min). Furthermore, a resting period of 5–10 min was allowed, after which an equal volume of sterile water and dichloromethane were added, thoroughly mixed, and subjected to another round of centrifugation at room temperature (13000 rpm for 3 min). Following the delamination of the liquid, flavonoids were found at the top, whereas carotenoids settled at the bottom.

### Content of total carotenoids

Due to their inherent instability and susceptibility to decomposition, carotenoids require extraction procedures conducted in a semi-dark environment. Carotenoids are soluble in ethanol, ether, and acetone, but insoluble in water, thereby enabling extraction with polar solvents such as ethanol and acetone (*Britton, 1995*; *Kiokias, Proestos & Varzakas, 2015*). Powdered floral organs were used as experimental materials. First, 0.1 g of the sample was accurately weighed and transferred into a 15–ml centrifuge tube, followed by the addition of 2 ml ethanol and thorough mixing. Centrifugation at 12000 rpm for 10 min was performed, and the resulting supernatant was carefully collected and transferred to a clean centrifuge tube. This process was repeated until the residue became colorless. Subsequently, the supernatant was merged, and its absorbance was measured at wavelengths of 665 nm, 649 nm, and 470 nm, with ethanol as the blank control. The calculation formulas for determining carotenoids are as follows:

$$\text{carotenoid concentration (mg/L)} : Cx = (1000A470 - 2.05Ca - 114.8Cb)/245 \quad (1)$$

where A470 represents the absorbance value of the ethanol solution containing carotenoids at 470 nm wavelength, Ca represents the content of chlorophyll a (Ca $= 13.95A_{665}$ - $6.88A_{649}$), and Cb represents the content of chlorophyll b (Cb $= 24.96A_{649}$ - $7.32A_{665}$). (2) carotenoid content (mg/g) $=$ (carotenoid concentration $\times$ extract volume $\times$ dilution multiple)/sample mass.

### HPLC analysis of carotenoid content in P. armeniacum

In this study, the HPLC analysis was conducted at Sanshu Biotechnology to quantify the carotenoid content in various flowering stages of *P. armeniacum*. The procedure involved extracting 0.1−0.5 g of ground samples with two mL of anhydrous ethanol containing 0.1% BHT (butylated hydroxytoluene), followed by treatment with 100 μL of 80% w/v potassium hydroxide solution. After heating in an 80 °C water bath and subsequent centrifugation, the extracts were dried under a gentle nitrogen flow at 30 °C and re-dissolved in 0.2 mL methanol for analysis. Chromatographic separation was achieved using a Thermo DGLC dual ternary UHPLC system, equipped with a DAD detector and a YMC carotenoid S-3 μm (150×4.6 mm) column. The analysis conditions included a column temperature of 40 °C, a detection wavelength of 450 nm, and a flow rate of 1.0mL/min using methanol (A) and a

mixture of methanol: MTBE: water (20:75:5 v/v/v, B) as mobile phases in a gradient elution program. The targeted carotenoids, including zeaxanthin, $\beta$-cryptoxanthin, lutein, $\beta$-carotene, $\alpha$-carotene, lycopene, capsanthin, violaxanthin, and neoxanthin, were quantified using an external calibration method, processed with Chromeleon 7 software.

### Extraction and quality detection of RNA

RNA was extracted from *P. armeniacum* flower tissue using Tiangen RNA Plant Plus kit (Tiangen Biotech Co., Beijing, China). The concentration, purity, and integrity of the extracted RNA were evaluated using agarose gel electrophoresis, Nanodrop, Agilent 2100 Bioanalyzer, and a Qubit Fluorometer. Only RNA samples that met the quality criteria were used to construct the transcriptome database. Full-length transcriptome data were generated by Novogene Co., Ltd. (Beijing, China), while the general transcriptome sequencing project was conducted and analyzed by BMKCloud Co., Ltd. (Beijing, China).

### Determination of full-length transcriptome

An equal mixture of the aforementioned qualified RNA samples was utilized for subsequent analysis of the full-length transcriptional library. The library was sequenced using the PacBio Sequel high-throughput sequencing platform, based on single-molecule real-time sequencing (SMRT) technology. The SMRTlink v6.0 software was employed to control the raw data and eliminate adapter sequences and low-quality reads. Following the CCS and ICE algorithms, the sequencing complex was used to cycle the same fragment through the circular library molecule to eliminate the base-read error rate, resulting in a polished consensus sequence for subsequent analysis. The polished consensus sequence was enhanced for sequencing accuracy through correction using LoRDEC software. CD-HIT software was employed for sequence comparison clustering to remove redundant and similar sequences while also generating the length frequency distribution of transcripts before and after redundancy removal (*Grabherr et al., 2011*; *Liu et al., 2023*).

### Sequencing and comparison of ordinary transcriptome

The corrected consensus sequence was de-redundant using CD-HIT software, resulting in a non-redundant transcript, referred to as the gene reference sequence (ref). The clean reads that passed the quality control were compared to the reference sequence using the kallisto algorithm for direct reading segment count (*Du et al., 2020*). Subsequently, count values for gene expression were determined. Finally, excluding the effects of sequencing depth and gene length on count values, a gene expression matrix composed of FPKM and TPM values was generated (*Zhou et al., 2022*).

### Screening of differentially expressed genes in carotenoid synthesis

To obtain comprehensive gene function information, the reference sequence was subjected to gene function annotation using multiple databases, including NCBI–nr, NCBI–nt, Pfam, EuKaryotic Orthologous Groups (KOG), Swiss-Prot, Kyoto Encyclopedia of Genes and Genomes (KEGG), and Gene Ontology (GO). The annotated information was used to identify the key structural genes involved in the carotenoid biosynthesis pathway. Differential expression analysis of carotenoid functional genes between groups was

**Table 1  Primer sequences and product lengths of genes used in RT-qPCR experiments.**

| Gene | Primer sequence | Product length |
|---|---|---|
| actin2 | F: GGGTGGCAGTTCGCTCTTCC | 132 |
|  | R: GCAGCACCTCTCCAGGCATC |  |
| i0c928 | F: TCAGGCGAGGAAGGTGAGAAGTG | 180 |
|  | R: CTGCCCACGGTAACAACATTTGAAG |  |
| i2c21133 | F: CGGCAATGCTGGTGAGGAGATTC | 126 |
|  | R: GCCTGGTTCTGTGCGTTACTGTC |  |
| i1c11866 | F: CGCCGACAAGTTGCCAACAATATC | 96 |
|  | R: CCTCCGCAGCCAGTGACAAAC |  |
| i0c7956 | F: CACCCTCTAGAAGCCCTACG | 131 |
|  | R: AGCGCTGGATAGAATCTGCT |  |
| i1c23264 | F: TTGCCCGAGATAGTGGTACC | 127 |
|  | R: TTGACCCGGGAAACAGCTAT |  |

conducted by DEG-Seq2 software (*Love, Huber & Anders, 2014*), with FPKM data applied for differential expression calculation. The screening criteria for significant differential expression of genes were as follows: $|\text{logFC}| \geq 1$ and Padj $\leq 0.05$. After screening, the genes were enriched using GO and KEGG through TBtools to ensure the reliability of the gene set screening (*Chen et al., 2020*).

### Prediction and screening of transcription factors

Transcription factors (TFs) represent a class of protein molecules that regulate the expression of target genes in specific temporal and spatial contexts. To identify plant transcription factors, iTAK software was employed for prediction, followed by screening the predicted genes for differential expression using the same criteria as carotenoid functional genes (*Zheng et al., 2016*). Subsequently, the differentially expressed genes from each group were integrated, and STEM software was utilized to enrich the differentially expressed transcription factors. By performing BLAST alignment with homologous genes in *Arabidopsis thaliana*, the key carotenoid regulatory transcription factors, including *MYB305*, *RCP1*, and *RCP2*, were identified. The Pearson algorithm was used to investigate the correlation between transcription factors with more than 50% homology and differentially expressed genes related to carotenoid synthesis ($P < 0.05$).

### RT-qPCR analysis of genes related to carotenoid synthesis

RNA extracted from the aforementioned samples was reverse transcribed into cDNA for RT-qPCR verification using the GoScript reverse transcription system (Promega, Madison, WI, USA). RT-qPCR was conducted using the SYBR premix ExTaq kit (Takara, Dalian, China) on an ABI 7500 real-time fluorescence quantitative PCR system (Applied Biosystems, Waltham, MA, USA). In this study, we selected five genes related to carotenoid synthesis to complete fluorescence quantitative analysis, and the primers used are shown in Table 1. The expression level was calculated as $2^{-\Delta\Delta Ct}$ and normalized to the Ct value of *P. armeniacum* actin2 (*Chang et al., 2016*; *Fang et al., 2020*).

**Table 2  Scale of total carotenoid content in P. armeniacum floral organs in different stages (three repeats per group).**

| Simple | Stage | Content (mg/100 g) | Average content (mg/100 g) | Standard deviation |
|--------|-------|--------------------|-----------------------------|---------------------|
| L1 | Bud | 7.446 | | |
| L2 | Bud | 5.022 | 6.370 | 1.236 |
| L3 | Bud | 6.656 | | |
| S1 | initial flowering stage | 9.991 | | |
| S2 | initial flowering stage | 10.208 | 10.671 | 0.995 |
| S3 | initial flowering stage | 11.813 | | |
| H1 | full flowering stage | 12.007 | | |
| H2 | full flowering stage | 13.367 | 11.858 | 1.589 |
| H3 | full flowering stage | 10.200 | | |

## RESULTS

### Analysis of flower color components in *P. armeniacum*

Through the application of the chemical reagent delamination experiment, the reaction results were observed for *P. armeniacum* flowers at different stages. The result of this step is very important. Yellow was concentrated in the lower layer, indicating that the yellow color of *P. armeniacum* is mainly caused by carotenoids, which lays the foundation for the following experiments (carotenoids exhibit higher solubility in non-polar solvents, whereas flavonoids display lower solubility in non-polar solvents. Following the aforementioned experiment, carotenoids will be located in the lower dichloromethane phase, while flavonoids will reside in the upper aqueous phase) (Fig. 1).

In the three stages, the lower layer of the solution in the bud stage appeared as a green solution, whereas the upper layer remained colorless and transparent. As flowering was initiated, the lower solution was yellow, while the upper layer was slightly yellow. In the full flowering stage, the yellow pigments were also concentrated in the lower layer, whereas the upper layer exhibited a slight orange-red tint with transparency. Quantitative assessment of carotenoids revealed a significant increase in their overall content throughout the flowering process of *P. armeniacum*. The most rapid growth rate was observed during the flower bud stage, although the growth rate gradually slowed from the initial flowering stage to the full flowering stage (Table 2).

### Determination of carotenoid content in *P. armeniacum* flower by HPLC

In this study, we scrutinized the changes in carotenoid content across different developmental stages of *P. armeniacum* flowers. High-performance liquid chromatography (HPLC) was employed to quantify the concentrations of major carotenoids. Our results reveal that specific carotenoids significantly increase and are closely tied to the floral color changes (Table 3).

Zeaxanthin, a key yellow pigment, showed the most significant increase, from 2.27 μg/100 mg in the bud stage to 3.91 μg/100 mg in the fully bloomed stage, an amplification of approximately 72%. Xanthophyll, another essential yellow pigment, also exhibited

**Table 3  Results of HPLC determination of carotenoid content (carotenoids not detected in all samples have been removed).**

| sample | Violaxanthin ug/100 mg | Xanthophyll ug/100 mg | Zeaxanthin ug/100 mg | β-Cryptoxanthin ug/100 mg | α-Carotene ug/100 mg | β-Carotene ug/100 mg |
|---|---|---|---|---|---|---|
| L | 0.241 ± 0.038 | 2.292± 0.094 | 2.272 ± 0.057 | 0.537 ± 0.037 | 0.006 ± 0.01 | 0.181 ± 0.007 |
| S | 0.508 ± 0.037 | 3.26 ± 0.211 | 3.429 ± 0.301 | 1.086 ± 0.103 | 0 | 0.158 ± 0.021 |
| H | 0.608 ± 0.009 | 3.007± 0.089 | 3.908 ± 0.081 | 0.947 ± 0.075 | 0 | 0.113 ± 0.017 |

a notable increase, from 2.29 µg/100 mg to 3.01 µg/100 mg, marking a 31% rise. Moreover, β-cryptoxanthin and violaxanthin concentrations showed remarkable increases. β-Cryptoxanthin rose by about 75%, from 0.54 µg/100 mg to 0.95 µg/100 mg, and Violaxanthin experienced a 154% increase from 0.24 µg/100 mg to 0.61 µg/100 mg.

In contrast, α-carotene and β-carotene demonstrated a slight decline during the blooming process, indicating their less significant role in the color change compared to the other carotenoids. In addition, the contents of neoxanthin, capsanthin and lycopene were 0 in all samples.

## Results of full-length transcriptome sequencing

The transcriptional statistical analysis of the corrected and de-redundant full-length transcriptome sequencing results indicated the successful acquisition of 75,216 valid transcripts. The longest transcript spanned 14,673 bases, whereas the shortest transcript comprised 200 bases, resulting in an average length of 3,633 bases. Transcripts were annotated across seven different databases (NCBI–nr, NCBI–nt, Pfam, KOG, Swiss-Prot, KEGG, and GO). The results indicated that a total of 25,317 genes were annotated in all the seven databases. Among the seven databases, NCBI–nr and KEGG exhibited the highest coverage, with 68,539 and 68,107 annotated genes, respectively (Fig. 2). Among them, the results of KEGG analysis were the most significant. The KEGG database provides a link between gene functions and metabolites. It draws generalized, detailed, and general reference metabolic pathways based on the existing knowledge.

Despite the abundance of studies on orchids, NCBI–nr database lacks the comprehensive inclusion of related data. Consequently, the annotation results of the transcriptome data obtained in this study demonstrated the absence of closely related species with high homology to *P. armeniacum*. In fact, the sequence homology between oil palm *(Elaeis guineensis)* and *P. armeniacum* was only 29%, which is the highest homology in the nr database. KEGG enrichment analysis highlighted the crucial role of metabolic pathways in flower color formation. Among these pathways, carbohydrate metabolism emerged as the most gene-rich pathway, with a total of 2,448 related genes. Through the prediction of transcription factors, 3,767 transcripts were identified as potential transcription factors. Among the annotated transcription factor families, the *SNF2* family exhibited the highest representation, with annotations observed in the transcript 384. In addition, *C3H, SET, PHD*, and other transcription factor protein families were also well represented. Regarding flower color formation, notable annotations included 118 for *bHLH*, 80 for *MYB*, and 163 for *MYB*-related transcription factors. In addition, we also conducted the development

 

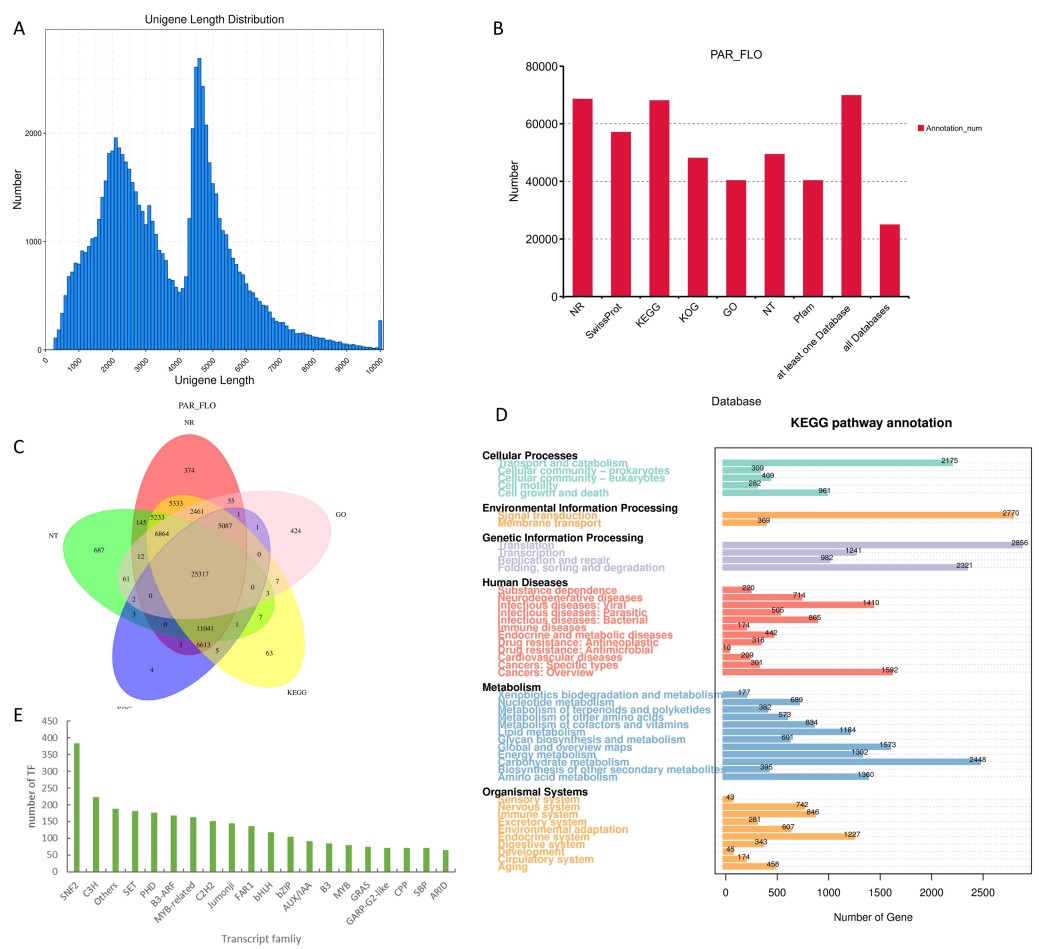

**Figure 2** **Results of full-length transcriptome sequencing.** (A) Distribution map of transcript length; (B) functional annotation of transcripts in different databases; (C) Venn diagram of annotated results in different databases; (D) pathway enrichment map of transcripts in the KEGG database; and (E) histogram of the top 20 transcription factors with the highest transcript counts.

of SSR tags and lncgene analysis of the transcript, with detailed data provided in the Supplementary Materials.

## Analysis of sequencing results of ordinary transcriptome

According to sequencing quality assessment, all 12 samples (bud stage (L), initial flowering stage (S), and full flowering stage (H) with four repeats in each group) were sequenced successfully. The data obtained exhibited excellent quality, with Q20 exceeding 97.39% and Q30 surpassing 92.87% (Table 4). The results of the kallisto comparison demonstrated a strong correlation of over 92% among the samples within each group. The gene expression levels in the samples (represented by the FPKM value) followed a normal distribution, and the differentially expressed genes displayed a balanced distribution among the groups (Fig. S1). These findings indicate that the transcriptome data are highly credible and are suitable for further analysis.

**Table 4** Evaluation form of quality-controlled data from the second-generation transcriptome sequencing.

| Sample ID | Obtained reads | Obtained base (bp) | Q20 (%) | Q30 (%) | GC (%) |
| --- | --- | --- | --- | --- | --- |
| L1 | 19,601,121 | 5,870,936,138 | 97.45 | 93.00 | 48.02 |
| L2 | 22,421,929 | 6,715,646,034 | 97.80 | 93.83 | 48.15 |
| L3 | 19,042,386 | 5,701,588,652 | 97.93 | 94.14 | 47.61 |
| L4 | 21,155,430 | 6,335,962,860 | 97.42 | 92.92 | 47.85 |
| S1 | 24,835,586 | 7,437,560,586 | 97.37 | 92.82 | 47.96 |
| S2 | 19,089,612 | 5,717,330,922 | 97.65 | 93.46 | 47.79 |
| S3 | 23,222,987 | 6,955,022,638 | 97.62 | 93.40 | 48.04 |
| S4 | 19,763,945 | 5,919,321,404 | 97.63 | 93.40 | 47.58 |
| H1 | 21,506,980 | 6,441,507,164 | 97.59 | 93.27 | 48.41 |
| H2 | 20,144,757 | 6,033,096,912 | 97.87 | 94.00 | 48.50 |
| H3 | 21,893,448 | 6,556,707,504 | 97.70 | 93.57 | 48.24 |
| H4 | 20,331,993 | 6,089,256,362 | 97.39 | 92.87 | 48.17 |

## Analysis of gene expression of transcriptome

Differential gene expression analysis within the transcriptional group was performed using DEseq2. The results indicated a total of 7,835 differentially expressed genes between L and H, with 3,856 genes up-regulated and 3,979 genes down-regulated. Similarly, a total of 9,604 genes exhibited differential expression between L and S, with 4,412 genes significantly up-regulated and 5,192 genes significantly down-regulated. Furthermore, 8,886 genes demonstrated differential expression between S and H, of which 4,598 genes were significantly up-regulated and 4,288 genes were significantly down-regulated. To comprehensively investigate the differential expression patterns, the differentially expressed genes across all groups were collected, yielding 14,786 differentially expressed genes.

We performed functional enrichment of all differentially expressed genes using GO and KEGG pathways to understand the genetic basis of its distinctive yellow coloration. The GO enrichment results revealed a significant concentration of genes involved in 'response to stimulus,' 'response to stress,' and 'oxidoreductase activity.' These findings suggest that the yellow coloration may be a result of the plant's adaptive response to environmental stimuli and stress, potentially involving complex redox processes that are integral to pigment formation. Furthermore, the KEGG enrichment analysis illuminated that among the most represented pathways were those associated with 'pattern recognition receptors,' 'photosynthesis antenna proteins,' and 'lipoic acid metabolism.' This predominance indicates a possible link between the recognition of environmental patterns, such as light intensity and quality, and the physiological adaptations that govern pigment biosynthesis. (Fig. 3).

## Exploration of differentially expressed carotenoid biosynthesis genes

The carotenoid biosynthesis pathway is extremely conserved, enabling us to effectively locate carotenoid biosynthesis-related genes according to functional gene annotation. In the analysis of full-length transcriptome sequencing data of *P. armeniacum*, we identified 113 genes associated with carotenoid biosynthesis. (Carotenoid biosynthesis genes can

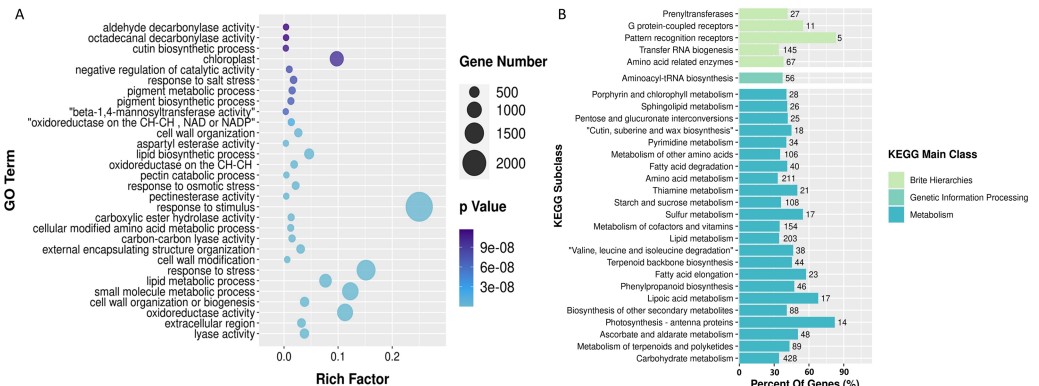

**Figure 3** GO and KEGG enrichment diagrams of differentially expressed genes among three groups of samples. (A) Bubble diagram depicting GO enrichment of differential genes; and (B) histogram displaying KEGG enrichment of differential genes, with the number at the end of the column representing the specific count of genes associated with the respective pathway.

be identified if there are carotenoid biosynthesis-related genes in the five databases). Subsequently, through differential gene analysis, 28 differentially expressed genes were linked to carotenoid biosynthesis.

The metabolic pathway relationships of all CBP genes and the expression levels of the differentially expressed genes are shown in Fig. 4. The *PDS* and *CYP97A3* gene families had the highest number of differentially expressed genes (five differentially expressed genes each), both of which are important intermediates in the carotenoid synthesis pathway. Enrichment analysis of 28 differentially expressed genes was conducted using GO and KEGG. GO enrichment analysis provided insights into the subcellular localization patterns of the differentially expressed functional genes. A significant proportion of these genes were predominantly localized in pigment cells, chloroplasts, and pigments. In contrast, the KEGG enrichment results highlighted the significant enrichment of differentially expressed genes within the carotenoid biosynthesis pathway.

Finally, we selected five genes related to carotenoid synthesis (including functional genes and transcription factors) to complete RT-qPCR, and the results were consistent with the transcriptome data (Fig. 4), indicating the reliability of the data

## Exploration of differentially expressed transcription factors

By comparing the predicted transcription factors with the results of the differential expression analysis, we identified a total of 877 differentially expressed transcription factors. To gain further insight into their expression patterns, STEM expression trend enrichment analysis was conducted. This suggested two highly enriched expression trends. The red trend indicates the largest number of differentially expressed genes, including *SNF2*, *C3H*, and *PDH* families. This finding was consistent with the transcription factor prediction data, which indicated these families as having the highest number of members. The green trend was characterized by the first three differential expressions of the *MYB*, *bZIP*, and *CH3* transcription factor families. Among all transcription factors, *MYB305*,

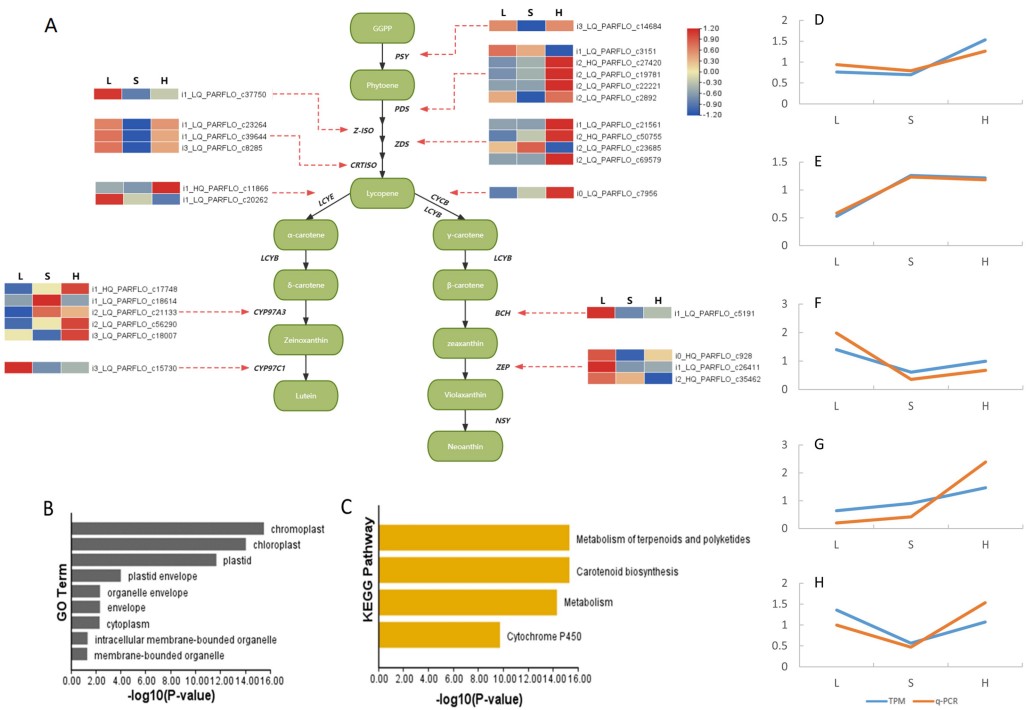

**Figure 4 Differentially expressed genes involved in carotenoid synthesis and their qPCR results.** (A) Heat map of differentially expressed genes in the carotenoid biosynthesis pathway. Green box represents intermediate compounds, and base is denoted by black and bold font. (B) Histogram presenting the GO enrichment results of subcellular localization for the differentially ex-pressed genes. (C) Histogram displaying the KEGG pathway enrichment results for the afore-mentioned differentially expressed genes. (D-H) qPCR results of genes related to carotenoid synthesis. These genes are i1c11866 (D), i2c21133 (E), i0c928 (F), i0c7956 (G) and i1c23264 (H). The TPM values of all genes and the relative expression values obtained by qPCR were divided by the mean of the data set to remove the dimensions between the data.

*RCP1*, and *RCP2* have been proven to be crucial positive regulators of carotenoid synthesis in floral organs. Through homology comparison and correlation analysis, 20 transcription factors were identified, including five *MYB305*, two *RCP1*, and 12 *RCP2*, which were strongly related to the expression of carotenoid synthesis genes (Fig. 5).

# DISCUSSION

*P. armeniacum* is a rare species within the *Paphiopedilum* genus, unlike *Phalaenopsis* orchids, which have been determined to use anthocyanins as the main chromogenic factor (*Liang et al., 2020*), few studies have been conducted on the reasons for the unique flower coloration of *P. armeniacum*. In this study, we focused on three developmental stages of *P. armeniacum*. By subjecting flower pigments to a series of separation steps, we isolated a distinct lower layer that exhibited a vivid yellow hue. This outcome was consistent with the well-documented higher solubility of carotenoids in non-polar solvents. As a result, our findings strongly affirmed the pivotal role of carotenoids in shaping the coloration of *P. armeniacum* flowers. We identified carotenoids as the primary contributors to the striking coloration of the *P. armeniacum* flowers. This result is similar to the results of Wang et al.'s

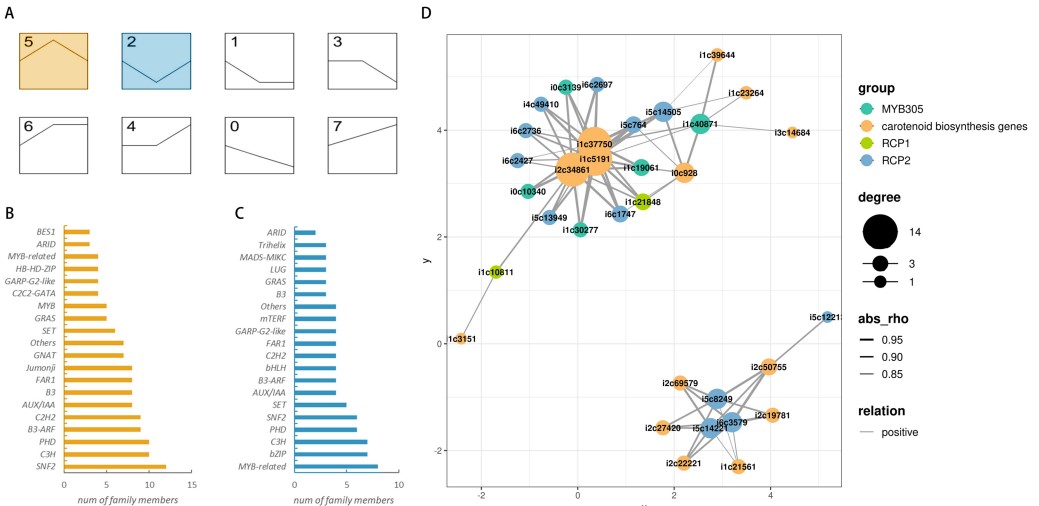

**Figure 5 Enrichment results of expression trends for differentially expressed transcription factors and the correlation network diagram of key transcription factors and carotenoid synthesis genes.** (A) Enrichment results of differentially expressed transcription factors for expression trends. Colored trends represent significant enrichment. The number in the upper left corner of each box represents the trend number and is unrelated to internal enrichment genes. (B) Distribution of transcription factors within trend 5. (C) Distribution of transcription factors within trend 2. (D) Network diagram of correlation between *MYB305*, *RCP1*, *RCP2* transcription factors and carotenoid biosynthesis genes.

study on the *Cymbidium* orchid, who suggested that the yellow *variety* of the *Cymbidium* orchid is mainly caused by carotenoids, while the rest of the color variety is caused by anthocyanins (*Wang et al., 2014a*; *Wang et al., 2014b*).

Subsequently, we quantified the carotenoid content in the flower tissues at various stages using absorbance measurements. The results revealed a rapid increase in the carotenoid content o from the flower bud to the early flowering stage. However, this increases stabilized during the later stage, which indicated substantial expression of carotenoid biosynthesis-related genes during the green flower bud stage. This may be because of the rapid increase in flower organs after the initial flowering stage. As floral organs grew rapidly, the synthesis rate of carotenoids per flower remained constant, resulting in a slower increase in unit content. Interestingly, we observed that larger flowers tended to demonstrate lighter yellow hues, which may also support this point.

To gain further insights into the accumulation characteristics of carotenoids during the flowering process of *P. armeniacum*, we conducted carotenoid determinations at different flowering stages using high-performance liquid chromatography (HPLC). The results revealed that zeaxanthin exhibited the most significant increase during the flowering process, followed by lutein. Both of these carotenoids are vital contributors to the yellow coloration observed in the flowers (*Kevan, Chittka & Dyer, 2001*; *Gitelson, Merzlyak & Chivkunova, 2002*). Additionally, although present in lower concentrations, $\beta$-carotene and violaxanthin concentrations also exhibited notable elevations. While these changes may not be as pronounced as those observed for zeaxanthin and lutein, they contribute to

the overall pigmentation of the flowers, highlighting the intricate interplay of carotenoids in coloration.

Transcriptomes offer a cost-effective and rapid approach for acquiring a vast number of single-gene sequences for organisms that lack reference sequences (*Lowe et al., 2017*). For example, in a recent study, *Ahmad et al. (2023)* successfully mined the chromogenic gene testa in peanuts using comparative transcriptomes. In this study, we conducted a comprehensive analysis and comparison of both full-length and common transcriptome data from *P. armeniacum*. During the sequencing process, the obtained data were high quality and met the criteria for subsequent analysis. In the common transcriptome sequencing data, the correlation between replicate samples within each group exceeded 92%, indicating excellent repeatability and ensuring the accuracy of our analyses.

During this analysis, we identified 28 differentially expressed carotenoid biosynthesis pathway (CBP) genes. Among these, the *BCH, ZEP*, and *CYP91C1* genes, crucial for the biosynthesis of carotenoids, were abundantly expressed at the flower bud stage, indicating their pivotal role in the early accumulation of carotenoids, which are vital for the yellow pigmentation in *P. armeniacum* flowers. Specifically, the *BCH* gene encodes beta-carotene hydroxylase, which is essential for the conversion of beta-carotene to zeaxanthin, a process that contributes to the yellow coloration (*Britton, 2008*). *ZEP* encodes zeaxanthin epoxidase, which catalyzes the epoxidation of zeaxanthin to violaxanthin, a precursor for abscisic acid synthesis that may influence flower maturation and pigmentation (*Ruiz-Sola & Rodríguez-Concepción, 2012*). *CYP97C1*, a member of the cytochrome P450 family, also participates in the hydroxylation of carotenoids, enhancing the diversity of carotenoid-derived pigments in the petals (*Kim & DellaPenna, 2006*).

Conversely, the upstream genes *PDS* and *ZDS* showed pronounced expression during the late flowering stage, potentially contributing to sustained carotenoid production. The *PDS* gene encodes phytoene desaturase, a key enzyme in the desaturation steps from phytoene to lycopene, while *ZDS*, the Zeta-carotene desaturase, furthers this desaturation process. Their higher activity towards the late flowering stage might be compensating for reduced carotenoid degradation as the flower matures, maintaining the intensity of the yellow hue (*Qin et al., 2007*). Additionally, *CRISTO* and *PSY* exhibited elevated expression levels in the early and late flowering stages, with a notable dip in the middle stage. This pattern suggests a regulatory mechanism that balances carotenoid synthesis with flower development stages. In particular, *PSY*, encoding phytoene synthase, catalyzes the first committed step in the carotenoid biosynthesis pathway and is often regarded as a rate-limiting step in the production of carotenoids, directly impacting the coloration intensity (*Welsch et al., 2000*).

The results from GO and KEGG analyses confirmed that all identified CBP genes are indeed localized in chromoplasts and chloroplasts and are integral to the carotenoid synthesis pathway. This localization underscores the dual role of these organelles in both photosynthesis and pigment accumulation for flower coloration. Our fluorescence quantitative analysis of selected carotenoid synthesis genes corroborated the transcriptome expression results, reinforcing the reliability of transcriptome sequencing in revealing the molecular underpinnings of flower pigmentation in this experiment.

Analysis of the expression trends of transcription factors revealed that the majority of transcription factors exhibited differential expression patterns at the beginning of flowering. This indicates the crucial role of *MYB305*, *RCP1*, and *RCP2* as significant positive transcription factors involved in carotenoid synthesis during the flowering process. In this study, a comprehensive analysis combining homology and correlation analyses identified 19 transcription factors (five *MYB305*, two *RCP1*, and 12 *RCP2*) that were highly associated with carotenoid synthesis during flowering. Interestingly, through correlation analysis, we can easily find three *RCP2* (i6_LQ_PARFLO_c3579 and i5_LQ_PARFLO_c14221) were highly positively correlated with only 6 carotenoid synthesis genes belonging to the *ZDS* or *PDS* families. This result highlights the distinctive regulatory role of the above three *RCP2* genes in the regulation of *ZDS* and *PDS* families, which is worthy of further investigation and exploration.

## CONCLUSIONS

*P. armeniacum*, an endemic plant of significant conservation value in China, is renowned for its rare apricot yellow flowers. In this study, we confirmed that carotenoids are the main pigments responsible for coloration during flowering of *P. armeniacum*, and explored the accumulation characteristics of carotenoids during flowering by extraction colorimetry and HPLC. By integrating the results of full-length and common transcriptome sequencing, we identified the key genes associated with carotenoid biosynthesis during the flowering process of *P. armeniacum*. Finally, functional annotation revealed 28 differentially expressed functional genes related to carotenoid synthesis, which were further validated by fluorescence quantitative analysis of the selected genes. Moreover, through the prediction of transcription factors, 19 transcription factors that exhibited a positive correlation with carotenoid synthesis during flowering were identified. Notably, three *RCP2* genes emerged as important regulatory factors regulating the *ZDS* and *PDS* gene families, which warrants further study. This study sheds light on the fundamental mechanism underlying color formation during the flowering process of *P. armeniacum*, which has significant implications for unraveling the mechanism governing carotenoid accumulation in the flowering process of *P. armeniacum*.

## ACKNOWLEDGEMENTS

I would like to express my gratitude to Professor Tao Hu for his great support on my project. Thanks to his guidance and help, I was able to complete my work. I also want to thank the research team for their collaboration and help in gathering data for my research project. Finally, I would like to thank MJEditor for linguistic assistance during the preparation of this manuscript and thank sanshu biotechnology for their help in determining carotenoids by HPLC.

### Funding

This research was supported by ICBR Fundamental Research Funds, grant nos. 1632020001, 1632021005, and 1632021018. The funders had no role in study design, data collection and analysis, decision to publish, or preparation of the manuscript.

### Grant Disclosures

The following grant information was disclosed by the authors:
ICBR Fundamental Research Funds: 1632020001, 1632021005, 1632021018.

### Competing Interests

The authors declare there are no competing interests.

### Author Contributions

- Yiwei Bai conceived and designed the experiments, performed the experiments, analyzed the data, prepared figures and/or tables, and approved the final draft.
- Jiping Ma conceived and designed the experiments, analyzed the data, prepared figures and/or tables, and approved the final draft.
- Yanjun Ma conceived and designed the experiments, prepared figures and/or tables, and approved the final draft.
- Yanting Chang conceived and designed the experiments, authored or reviewed drafts of the article, and approved the final draft.
- Wenbo Zhang conceived and designed the experiments, authored or reviewed drafts of the article, and approved the final draft.
- Yayun Deng conceived and designed the experiments, authored or reviewed drafts of the article, and approved the final draft.
- Na Zhang performed the experiments, authored or reviewed drafts of the article, and approved the final draft.
- Xue Zhang performed the experiments, authored or reviewed drafts of the article, and approved the final draft.
- Keke Fan performed the experiments, authored or reviewed drafts of the article, and approved the final draft.
- Xiaomeng Hu performed the experiments, authored or reviewed drafts of the article, and approved the final draft.
- Shuhua Wang performed the experiments, authored or reviewed drafts of the article, and approved the final draft.
- Zehui Jiang conceived and designed the experiments, authored or reviewed drafts of the article, and approved the final draft.
- Tao Hu conceived and designed the experiments, authored or reviewed drafts of the article, and approved the final draft.

### Data Availability

The transcriptome sequencing data are available at NCBI SRA: PRJNA977114.

## Supplemental Information

Supplemental information for this article can be found online at http://dx.doi.org/10.7717/peerj.16914#supplemental-information.

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
