# Peer review of "Color components determination and full-length comparative transcriptomic analyses reveal the potential mechanism of carotenoid synthesis during Paphiopedilum armeniacum flowering"

_PeerJ, doi:10.7717/peerj.16914_

## Round 0.1 · original submission · Major Revisions

Thank you for submitting your work to PeerJ. The manuscript was reviewed by 5 expert reviewers and read by me. The reviewers were generally supportive of the manuscript, but made a few suggestions to improve the impact of the manuscript.

In particular, the qRT-PCR verification of RNAseq data needs to be extended, with focus on the discussed genes and the results plotted against each other (TPM vs. dCt).

The HPLC analysis of carotenoids as suggested by rev. 5 would indeed be valuable but it may be beyond the scope of this study.

In addition to the reviewers' comments, Figure 2d and 3 are basic descriptions of data and can be put in supplement and in Fig. 4 it is not clear what is compared with what, i.e. what kind of DEGs you analyse.

**Language Note:** PeerJ staff have identified that the English language needs to be improved. When you prepare your next revision, please either (i) have a colleague who is proficient in English and familiar with the subject matter review your manuscript, or (ii) contact a professional editing service to review your manuscript. PeerJ can provide language editing services - you can contact us at [email protected] for pricing (be sure to provide your manuscript number and title). – PeerJ Staff

Reviewer 1 ·

Basic reporting

The reference format for this article still needs to be standardized, such as Line 477.

Experimental design

no comment

Validity of the findings

no comment

Additional comments

In the discussion, the authors need to analyze the key genes of carotenoid synthesis and related transcription factors in more detail, explore the relationship between them, and construct the regulatory network of carotenoid synthesis in P. armeniacum.

Reviewer 2 ·

Basic reporting

Paphiopedilum armeniacum is an important commercial flower, and the changes in flower color during its development are mainly caused by changes in carotenoid content. They integrated both third-generation full-length transcriptome sequencing and second-generation high-throughput transcriptome sequencing, to comprehensively explore the molecular mechanism involved in coloration.

Experimental design

The experimental design is basically correct, and the data is detailed and complete.

Validity of the findings

no comment

Additional comments

In Figure 5d, qRT PCR results need to be redrawn into a line chart for better display.

Reviewer 3 ·

Basic reporting

The authors described that they executed RT-qPCR analysis on the selected carotenoid synthesis genes, but I cannot find this result in the text.

Experimental design

no comment

Validity of the findings

no comment

Additional comments

1. Line 162, “2.2.5. ” rather than “4.2.5.” And this title should be bold.
2. Line 175, Check that this reference is citing in the correct place. I think it should be cited at the end of the sentence.
3. Line 196, “-ΔΔCt ” should be superscript.
4. Line 219, “The longest transcript spanned 14,673 bases, whereas the shortest comprised 14,673 clips ”. Is the shortest also 14,673?
5. Lines 226-229, This paragraph doesn't belong here.
6. Lines 253-254, This description is no need to cite references.
7. Line 219 and Line 3301: “Arabidopsis thaliana ” and “P. armeniacum ” should be italic.

·

Basic reporting

The paper titled "Color Components Determination and Full-Length Comparative Transcriptomic Analyses Reveal the Potential Mechanism of Carotenoid Synthesis During Paphiopedilum armeniacum Flowering" presents a comprehensive investigation into the intriguing process of flower coloration in Paphiopedilum armeniacum, a native ornamental plant in China known for its striking yellow blossoms. The study employs a dual-pronged approach, combining physicochemical analyses and advanced transcriptome sequencing, to shed light on the mechanisms underlying the vibrant hues of these flowers. The strength of this research lies in its rigorous methodology, which successfully identifies carotenoids as the primary pigments responsible for the flower colors, pinpoints differentially expressed carotenoid biosynthesis genes, and highlights potential regulators of this process. However, while the study is comprehensive, several points still need attention before it can be considered for publication. Following are my specific comments.

Experimental design

Comments for authors:
Abstract
1. Line 27-29: The sentences are not clearly structured. I suggest to revise these sentences with more appropriate words which simply convey the message.
Introduction
2. Lines 39-53: The reference to past awards and breeding potential of the P. armeniacum is informative, but it might be helpful to briefly mention how the flower color is crucial in these aspects.
3. (Lines 48-51): It would be beneficial to provide more specific details on how flower color is crucial for breeding endeavors. This would help readers understand the significance better.

4. Lines 54-72: Relevant references to specific plant species and their pigments are beneficial for context. I recommend the following references should be cited to enrich the context appropriately:
https://doi.org/10.1186/s13568-019-0854-x
https://doi.org/10.3390/molecules28073205

5. Lines 83-92: The mention of MYB305, RCP1, and RCP2 as regulators of carotenoid synthesis is informative, although a brief explanation of their functions would be beneficial.

Validity of the findings

Methods
6. (Lines 114-126): The method for differentiating between flavonoids and carotenoids based on chemical properties is described clearly. However, it might be helpful to mention why these specific developmental stages were chosen for analysis and how uniformity was ensured.

7. (Lines 127-143): The extraction method for carotenoids is well-explained, including the use of polar solvents and the calculation formula. Ensure clarity regarding "pigment concentration" in the carotenoid content formula. Is this a variable obtained from the experiment?

8. (Lines 151-161): The sequencing and analysis process for full-length transcriptome data is clear. However, it might be beneficial to briefly explain what CCS and ICE algorithms are and what they achieve. Specify why the PacBio Sequel platform was chosen for this analysis.

Results
9. (Lines 215-245): A brief explanation of the significance of the KEGG enrichment analysis results could be beneficial for readers who may not be familiar with this analysis.

10. In section 3.1. Analysis of Flower Color Components in P. armeniacum: The chemical reagent delamination experiment results clearly support the dominance of carotenoids in the yellow pigments. However, it would be beneficial to explain why this is a critical finding and how it contributes to the overall understanding of flower coloration in P. armeniacum.

11. In section 3.4. Analysis of Gene Expression of Transcriptome: The information about differentially expressed genes and their enrichment analysis is comprehensive. However, it might be helpful to briefly elaborate on the significance of the enriched pathways for the study's objectives.

12. In section 3.5. Exploration of Differentially Expressed Carotenoid Biosynthesis Genes: What specific role do the 28 differentially expressed carotenoid biosynthesis genes play in P. armeniacum's flower coloration? How do these genes correlate with the observed increase in carotenoid content during flowering?
Discussion

13. The identification of carotenoids as the primary contributors to the flower coloration of P. armeniacum is a significant finding. However, it would be helpful to briefly discuss how this finding compares to other orchid species or related research.

14. How do the findings in P. armeniacum regarding the dominance of carotenoids align with or differ from similar studies in orchids or other plant species?

15. The discussion of carotenoid content changes during flower development is informative. It helps understand the dynamics of carotenoid biosynthesis. It might be beneficial to include a visual representation or graph showing the changes in carotenoid content throughout the different stages.

16. The section effectively highlights the importance of transcriptome analysis in this study. However, I suggest the authors should discuss other transcriptome studies for the identification of DEGs specific to other functions. Following papers can be cited here:

https://doi.org/10.3390/genes12101492
https://doi.org/10.3390/genes13050841
https://doi.org/10.3390/ijms23126376
https://doi.org/10.1186/s12870-023-04041-0

Additional comments

17. Could you elaborate on the potential reasons for larger flowers having lighter yellow hues? Are there any hypotheses regarding this observation?

18. Could you discuss the specific functions and roles of BCH, ZEP, CYP91C1, PDS, and ZDS in carotenoid biosynthesis and their importance in the context of flower coloration?

Reviewer 5 ·

Basic reporting

The paper “Color components determination and full-Length comparative transcriptomic analyses reveal the potential mechanism of carotenoid synthesis during Paphiopedilum armeniacum flowering” by Bai et al. mainly identified some differentially expressed carotenoid biosynthesis genes throughout the flowering process and discovered some potential positive regulators involved in carotenoid synthesis. It is a very difficult that carotenoid biosynthesis genes and transcription factors can be involved in the process of biosynthesis during Paphiopedilum armeniacum flowering. The difficulty may also lie in the fact that some of the genes may be specific to only one tissue or one plant species. The study provides a basic information of physiological and molecular aspects about carotenoid synthesis during Paphiopedilum armeniacum flowering. Unfortunately, there are some key issues to be addressed.
1. Flower includes petals, pistils, and stigma tissues. The expression patterns of carotenoid biosynthesis genes and regulators are different in petals, pistils, and stigma tissues. Which tissue was experimental material?
2. During Paphiopedilum armeniacum flowering the content and composition of carotenoids should be determined using liquid chromatography mass spectrometry (LC-MS).
3. qRT-PCR analysis should be used to confirm the carotenoid biosynthesis genes and potential positive regulators.

Experimental design

1. Flower includes petals, pistils, and stigma tissues. The expression patterns of carotenoid biosynthesis genes and regulators are different in petals, pistils, and stigma tissues. Which tissue was experimental material?
2. During Paphiopedilum armeniacum flowering the content and composition of carotenoids should be determined using liquid chromatography mass spectrometry (LC-MS).
3. qRT-PCR analysis should be used to confirm the carotenoid biosynthesis genes and potential positive regulators.

Validity of the findings

no comment

Additional comments

no comment

---

## Round 0.2 · accepted · Accept

The revised manuscript sufficiently addressed all the reviewers´ comments and is now ready for publication.

Reviewer 3 ·

Basic reporting

no comment

Experimental design

no comment

Validity of the findings

no comment

Additional comments

The authors have revised the manuscript according to my suggestions and further improved the manuscript. A small number of English language errors require further scrutiny.

·

Basic reporting

Authors have improved the manuscript by following all the suggestions and recommendations.

Experimental design

Authors have added few qPCR experiments, modified the mutations according to suggestions, and presented the results in the form of line chart (TPM vs. dCt) after averaging the data. Now Figure 4 is providing much better information.

Validity of the findings

Article is now in much better format then the previous version.

Additional comments

This article can be accepted for publication in its current form.